# Impact of COVID-19 on older adults with cancer and their caregivers' cancer treatment experiences study: The ICE-OLD study

Cydney Low[1], Isabel Tejero[2], Nelly Toledano[1], Caroline Mariano[3], Shabbir Alibhai[4], Manon Lemonde[5], Kristen Haase[6‡], Martine Puts[1‡*]

1 Lawrence S. Bloomberg Faculty of Nursing, University of Toronto, Toronto, Ontario, Canada, 2 Geriatrics Department, Hospital del Mar, Barcelona, Spain, 3 BC Cancer Agency, Vancouver and Royal Columbian Hospital, New Westminster, British Columbia, Canada, 4 Department of Medicine and Institute of Health Policy, Management, and Evaluation, University Health Network and University of Toronto, Toronto, Ontario, Canada, 5 Faculty of Health Sciences, Ontario Tech University, Oshawa, Ontario, Canada, 6 School of Nursing, University of British Columbia, Vancouver, British Columbia, Canada

‡ KH and MP are joint last author on this work.
* martine.puts@utoronto.ca

**Data Availability Statement:** The participants of this study did not give written consent for their data to be shared publicly, so due to the sensitive nature of the research supporting data is not available. The Research ethics Committee in accordance with

## Abstract

The COVID-19 pandemic and health services impacts related to physical distancing posed many challenges for older adults with cancer. The goal of this study was to examine the impact of the pandemic on cancer treatment plans and cancer treatment experiences of older adults (ie, aged 65 years and older) and their caregiver' experiences of caring for older adults during the pandemic to highlight gaps in care experienced. In this multi-centre qualitative study guided by an interpretive descriptive research approach we interviewed older adults diagnosed with cancer and caregivers caring for them. Participants were recruited via cancer treatment centres in the provinces of British Columbia and Ontario (Vancouver and Toronto), Canada, and through an online ad sent out through patient advocacy organization newsletters. Interviews were recorded and transcribed verbatim and data were analyzed using an interpretive thematic analysis approach. A total of 27 individuals (17 older adults, 52.9% female; 10 caregivers, 90% female) participated in interviews lasting on average 45 minutes. Older adults with cancer described many impacts and pressures created by the pandemic on their cancer experiences, though they generally felt that the pandemic did not impact treatment decisions made and access to care. We grouped our findings into two main themes with their accompanying subthemes, related to: (1) alterations in the individual and dyadic cancer experience; and (2) navigating health and cancer systems during the pandemic. The additional stressors the pandemic placed on older adults during their treatment and decision-making process and their caregivers expose the need to create or avail additional supports for future disruptions in care.

the tri-council statement in Canada has not given permission to share the data. Due to the nature of the interviews, the data cannot be de-identified further to be able to share them anonymously. The contact information of the University Health Network is reb@uhnresearch.ca contact info+ 1 416-581-7849.

**Funding:** The author(s) received no specific funding for this work.

**Competing interests:** The authors have declared that no competing interests exist.

## Introduction

Cancer is the leading cause of morbidity and mortality in Canada and is responsible for 30% of all deaths [1]. Nearly half of all Canadians will develop cancer during their lifetime, and approximately one quarter of Canadians are expected to die from the disease [1]. The number of new cancer diagnoses and cancer-related deaths is steadily increasing due to the aging population [1]. Older adults also accounted for 17.5% of the over 3.9 million confirmed total number of COVID-19 infections, 64.2% of COVID-19-related hospitalizations, 59.7% of intensive care unit admissions, and 92.6% of COVID-19 deaths [1]. Thus older adults with cancer faced challenges related to their increased risk of infection and the potential for deleterious outcomes. During the early days of the pandemic, public health officials enacted physical distancing rules to reduce the spread of infection in many countries including Canada. Consequently, there were cancellations and delays in routine health check-ups, cancer screenings, cancer surgeries and diagnostic procedures, non-urgent cancer treatments, and follow-up appointments to ensure resources were available for COVID-19 cases [2–19]. Moreover, these health measures posed challenges for older adults with cancer including decreased physical activity; increased social isolation; impacted financial status; reduced access to home care and community supports; and uncertainty about the future [20–23].

Traditionally, the predominant model of care utilized by oncology teams in Canada is face-to-face outpatient consultation [24]; however, since the beginning of the pandemic, the use of telehealth to facilitate patient-provider communication increased significantly [25]. Recommendations from co-operative oncology groups early in the pandemic emphasized the need to minimize visits for in-person therapy, switches to oral therapy, or omission of cancer-directed therapy in some cases, particularly for vulnerable populations [26, 27]. Vulnerable populations in this context refers to those with multiple comorbidities, increased age, and low chance of survival when exposed to the excess risk from both COVID-19 and under-treatment of cancer [26].

As the pandemic stretched on, the use of telehealth applications such as email, virtual conferencing, and mobile applications utilized in oncology garnered considerable interest [28–32]. Personal experiences and preferences of cancer patients and clinicians has also been increasingly examined [33–39]. Previous qualitative studies conducted during the pandemic highlighted positive experiences such as increased frequency of patient-clinician interactions, improved workflow and job satisfaction, convenience, and access to medical professionals [40–44]. Negative experiences include limited physical consultations, communication barriers, technological difficulties, and impacted patient-clinician relations [41, 43–45]. Although there has been an up-tick in research on patient and clinician experiences related to telehealth and the impact of COVID-19 concluding that many patients were happy with telehealth due to less traveling.

Older adults may have different experiences using telehealth due to age-related changes in vision, hearing and cognition [46, 47]. In addition, many older adults rely on family caregivers to help them make treatment decisions and complete cancer treatments but this group is often neglected in research. Previous studies report that women diagnosed with breast cancer and hematological malignancies during the pandemic missed their family members during appointments, hospital admissions for cancer surgeries [34, 38, 48]. A study interviewing patients with cancer and their spouses showed that caregivers experienced more challenges in caring for their spouses due to inability to go into hospital and access information [49].

The purpose of this study was to generate an in-depth understanding of the cancer treatment and decision-making experiences of older adults with cancer and their caregivers during the COVID-19 pandemic. The findings of this study will be valuable to guide approaches to

care in future pandemics and in situations calling for augmented public health safety measures.

The research questions were:

1. How does the COVID-19 pandemic impact cancer treatment plans and decision-making for older adults with cancer and their caregivers?

2. What are the cancer treatment experiences of older adults with cancer and their caregivers during the COVID-19 pandemic?

3. What are caregivers' experiences of caring for older adults with cancer during the COVID-19 pandemic?

## Methods

We used an interpretive descriptive approach to guide the research process. As described by Thorne [50], this is a reflexive and inductive practice-oriented qualitative research methodology which aligns with the pragmatic approach of the research team [50]. This approach included interviews with individuals diagnosed with cancer and their caregivers, March 2021–August 2021.

Ethics approval was obtained from the University Health Network research ethics board (#20–5948), the University of Toronto research ethics board (#40167), and the University of British Columbia/BC Cancer Harmonized ethics review process (# H20-03957). Research in Canada is guided by the Tri-Council Policy Statement (https://ethics.gc.ca/eng/tcps2-eptc2_2018_introduction.html#1) and the International Council for Harmonisation of Technical Requirements for Pharmaceuticals for Human Use (ICH) Good Clinical Practice guideline. All older adults and caregivers provided written informed consent prior to the interviews. We used the Consolidated criteria for Reporting Qualitative research (COREQ) Checklist for reporting the study [51] (S1 Checklist).

### Sample

We used a maximum variation purposive sampling approach and participants were recruited from 2 hospitals (University Health Network, Toronto, ON and Royal Columbian Hospital, New Westminster, BC) and via advertisements in e-newsletter through a national patient advocacy group and an Ontario cancer support group. Eligible patients were those who were age 65 years or older; recently diagnosed with cancer ($\leq$ 12 months); able to participate in the interview; and complete the questionnaire; or a caregiver of an individual meeting the same criteria. Recruitment continued until there was repetition in themes which robustly addressed the research questions.

### Recruitment and data collection

Individuals in the hospital setting were first approached by either a physician or nurse in person or via telephone to ask if they were interested in hearing more about the study using convenience sampling. If they agreed to be contacted, the study coordinator contacted the participant to review consent, study procedures, and inclusion/exclusion criteria. An interview was arranged to be over MS Teams/Zoom or via telephone pending participant's preference. For participants whose caregiver agreed to be included in the study, the patient and caregiver could choose to do the interview together as a dyad or each an individual interview. Participants were interviewed using a topic guide developed by the multidisciplinary study team consisting of nurse researchers, a medical oncologist, geriatric oncologist and geriatricians based

on their experiences caring for older adults with cancer (see S1 File). Questions included (1) what they heard/knew about COVID-19; (2) challenges experienced in their cancer care; (3) perceived impact of public health measures such as physical distancing on their cancer experience and move to virtual care; and (4) effects on physical and mental health, and perceived supports and use of care. All participants were asked for suggestions to improve care for future patients during the pandemic. Interviews were conducted by experienced qualitative PhD prepared nurse faculty (KH and MP both female) and a geriatric oncology fellow (IT, female), audio recorded (telephone interviews) or video recorded (MS Teams/Zoom interviews), and transcribed verbatim. The participants and interviewers had no relationships prior to the interview. At the start of the interview, the interviewers reiterated the purpose of the study, asked permission for the audio recording and provided an opportunity for participants to ask questions. One dyad declined to have the interview audio recorded and for this interview the field notes were used in the analysis.

At the end of the interviews, a short survey was completed to obtain the sociodemographic characteristics of the participants (including age, sex, marital status) and medical history and cancer-and treatment-related information (approximate date of cancer diagnosis, self-reported cancer treatment received/planned and self-reported Charlson Comorbidity Index) to describe the sample [30]. In addition, caregivers were asked additional questions about how much time per week they provided care, type of caregiver (spouse, friend, child etc.) and since when.

## Analysis

We used an interpretive thematic analysis approach [50] aided by NVivo software. Five authors (all healthcare professionals with experience working with older adults with cancer) were directly involved in the thematic analysis (CL, IT, NT, MP, KH), they all coded all transcripts and then met on a regular basis to develop a coding framework and have discussions about fit and congruence across the developing themes and sub-themes. Three authors were trainees and relatively new to qualitative research (CL, IT and NT) and they were supervised by MP and KH who are experienced qualitative researchers. The analysis process involved four steps: initial reading for context and understanding; a second reading to inductively derive conceptual themes and assigning codes; analyzing developing codes, notes and grouping codes according to themes; and organizing the conceptual themes into an analytic structure. We addressed rigour by following Thorne's approach [50] with attention to coherence between the research question, methods, and analysis; keeping analytic memos and interpretations to inform analysis and support decision making; and ensuring that conclusions made are consistent with the research questions. We used descriptive statistics such as frequencies of the survey data to describe the participants.

## Results

### Description of the sample

A total of 27 individuals participated in the study and were interviewed between March 2021 and August 2021 (17 older adults with cancer and 10 caregivers). Eight caregivers and patients were interviewed together, and two caregivers were interviewed separately. Interviews lasted on average 45 minutes. Sixteen interviews were conducted via telephone and 2 by videoconference. More than half of the older adults participants (58.8%) were in the age range of 70–80 years old, see Table 1. Of the 17 older adults with cancer, nine identified as female (52.9%) and eight identified as male (47.1%).

More than half of the caregiver participants were in the age range 66–80 years old (60%), see Table 2. Nine identified as females (90.0%) and one identified as male (10.0%). Caregivers

**Table 1. Socio-demographic information for patients.**

| Characteristic | N = 17 | % |
|---|---|---|
| **Age** (years) | | |
| <70 | 3 | 17.6 |
| 70–75 | 4 | 23.5 |
| 76–80 | 6 | 35.3 |
| 81–85 | 3 | 17.6 |
| ≥86 | 1 | 5.9 |
| **Sex** | | |
| Male | 8 | 47.1 |
| Female | 9 | 52.9 |
| **First language** | | |
| English | 15 | 88.2 |
| Italian | 1 | 5.9 |
| Other | 1 | 5.9 |
| **Housing situation** | | |
| At home (house, condo, or apartment) | 17 | 100.0 |
| **Living situation** | | |
| Alone | 3 | 17.6 |
| Spouse | 11 | 64.7 |
| Child(ren) | 2 | 11.8 |
| Other | 1 | 5.9 |
| **Marital Status** | | |
| Married or living common law | 11 | 64.7 |
| Widow/widower | 4 | 23.5 |
| Separated or divorced | 1 | 5.9 |
| Single (never married) | 1 | 5.9 |
| **Education level** | | |
| 5 to 8 years (grade school to middle school) | 1 | 5.9 |
| 9 to 12 years (some or completed high school) | 5 | 29.4 |
| 13 years and more (some or completed college or university) | 11 | 64.7 |
| **Current cancer treatment (within the past 30 days)** | | |
| Surgery | 3 | 20.0 |
| Radiation | 3 | 20.0 |
| Chemotherapy/targeted therapy/immunotherapy | 9 | 60.0 |
| Hormone therapy | 5 | 33.3 |
| Other | 2 | 13.3 |
| **Treatment Intent as Reported by Patient** | | |
| Curative | 11 | 68.8 |
| Palliative | 5 | 31.3 |
| **Current co-morbidities** | | |
| Asthma, emphysema, chronic bronchitis, COPD | 3 | 17.6 |
| Arthritis or rheumatism | 7 | 41.2 |
| Diabetes | 2 | 11.8 |
| Digestive problems (ulcer, colitis, gallbladder disease, etc.) | 1 | 5.9 |
| Heart trouble (angina, congestive heart failure or coronary artery disease) | 5 | 29.4 |
| Kidney disease | 1 | 5.9 |
| Depression and/or anxiety | 1 | 5.9 |
| Other (n = 16) | 3 | 18.8 |

**Table 2. Socio-demographic information for caregivers.**

| Characteristic | N = 10 | % |
|---|---|---|
| **Age** (years) | | |
| 30–50 | 1 | 10.0 |
| 51–65 | 3 | 30.0 |
| 66–80 | 6 | 60.0 |
| **First language** | | |
| English | 9 | 90.0 |
| Other | 1 | 10.0 |
| **Sex** | | |
| Male | 1 | 10.0 |
| Female | 9 | 90.0 |
| **Living situation** | | |
| Spouse | 8 | 80.0 |
| Other | 2 | 20.0 |
| **Marital Status** | | |
| Married or living common law | 8 | 80.0 |
| Separated or divorced | 1 | 10.0 |
| Single (never married) | 1 | 10.0 |
| **Education level** | | |
| 9 to 12 years (some or completed high school) | 3 | 30.0 |
| 13 years and more (some or completed college or university) | 7 | 70.0 |
| **Current health problems** | | |
| Arthritis or rheumatism | 1 | 10.0 |
| Heart trouble (angina, congestive heart failure or coronary artery disease) | 1 | 10.0 |
| Other | 3 | 30.0 |
| **Duration of family member support (n = 9)** | | |
| <1 year | 2 | 22.2 |
| 1 to 5 years | 4 | 44.4 |
| >10 years | 3 | 33.3 |
| **Average amount of support hours provided per week (n = 9)** | | |
| <5 hours | 2 | 22.2 |
| 5 to 10 hours | 1 | 11.1 |
| 11 to 20 hours | 1 | 11.1 |
| >20 hours | 5 | 55.6 |

included a child, spouse, or friend of a patient. Most participants indicated that English was their first language and that they had attended College or University. Five patients started receiving cancer care prior to the pandemic; the other patients (n = 12) were diagnosed since the declaration of the pandemic.

**Results of qualitative study.** Through our thematic analysis we developed two major themes relating to older adults and caregivers' cancer treatment experiences during the COVID-19 pandemic: (1) alterations in the individual and dyadic cancer experience; and (2) navigating health and cancer systems in the midst of the pandemic.

Please see Fig 1 for a visualization of the main themes.

We noted that there were regional differences in experiences in patients recruited in British Columbia versus those recruited in Ontario. In Ontario, due to provincial pandemic restrictions which led to the longest lockdowns in the world [52], caregivers reported not being able

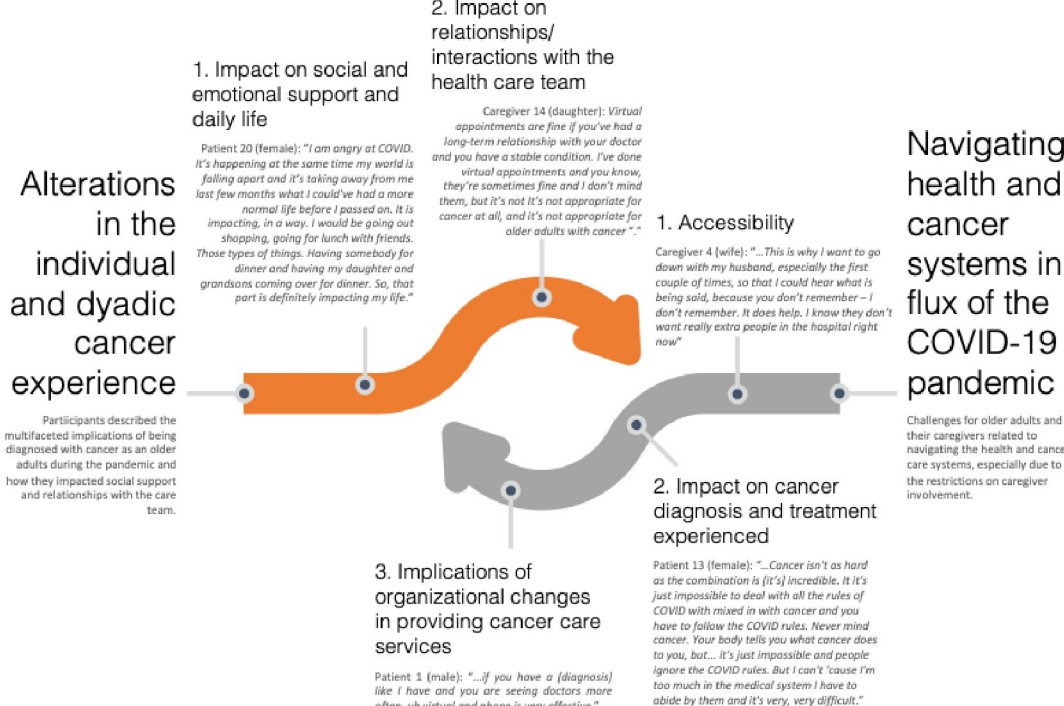

**Fig 1. Thematic visualization how the COVID-19 pandemic impacted cancer treatment plans and decision-making for older adults with cancer and their caregivers.**

to come to appointments and experiencing a lot of difficulty trying to visit their loved ones during hospital admission. Most older adults and their caregivers recruited in British Columbia did not describe being unable to bring their support person to the appointment with them or having their support person able to visit them in hospital when they were admitted.

Below is a description of the themes and their subthemes.

**Theme 1: Alterations in the individual and dyadic cancer experience.** Within this theme we identified two subthemes: 1) Impact on social and emotional support and daily life; and 2) Impact on relationships/ interactions with the health care team.

*Subtheme 1*: *Impact on social and emotional support and daily life*. Patients experienced isolation from family and friends due to provincial stay-at-home orders and restricted visitation policies enacted in health care settings including cancer centres and physician's offices. This caused significant negative impacts on patients' social support systems. Although some patients attempted to communicate with family and friends using virtual methods, such as Zoom or Facetime calls, the experience proved difficult and ineffective for many.

For study participants, the traditional stressors expected during cancer treatment were compounded by the COVID-19 pandemic. Older adults with cancer on active treatment were asked to self-isolate to reduce the risk of infection. Older adults who lived alone and/or who were self-isolating from their family members indicated missing the emotional support necessary to effectively cope with their recent cancer diagnosis or treatment side-effects. Several of the older adults with children indicated they only saw their children who lived in the same house, and that they were unable to see the children who lived elsewhere. Patients described missing the support and connection their children provided because of the restrictions of the pandemic. Additionally, participants identified that the burden of care often fell on the child they lived with.

Although many patients and caregivers reported being able to cope with the restrictions caused by the pandemic, some expressed negative psychological consequences accompanying restricted visitation policies and stay-at-home orders. Many older adults and caregivers understood the need for the restrictions and were coping well with the restrictions and adapted to visiting in a garden/park with family members and friends, etc. Particularly during admission to hospital for surgery or treatment, it was difficult for caregivers to be restricted from visiting their loved ones. Furthermore, many caregivers described feeling worried about the quality of care provided to their loved ones admitted to hospital because patients were isolated in their rooms, staffing shortages, and some noticing delirium in their loved ones. However, the pandemic lockdowns made it more convenient for caregivers to provide care for their loved ones due to the ability to work from home. Due to pandemic restrictions, caregivers were required to provide care on behalf of health care professionals, such as participating in exercises, daily vital monitoring, etc. In addition, other family members and friends were unable to assist in providing support to the older adult due to social distancing leading to all the burden falling on the primary caregiver. Please see below for exemplar quotes.

> Caregiver 14 (daughter): "*I'm all stressed. And my mum has zero emotional support and it's and it's all because of COVID. And it's terrible. And it, yes, it does affect our relationship because then when it resolves, she says to me, 'I wanted a daughter. I need a daughter. I needed someone to hold my hand. I needed someone to hug me. I needed someone to tell me I was going to be OK and you didn't do that for me. Why didn't you do that for me? Why weren't you there for me?*"

> Patient 1 (male): "*Because I live alone and because my immunity was so bad and this was going for a year for me, I hardly see anybody and if I do see anybody. . . I did not see anyone in my house for 6 months. So, it's pretty hard.*"

> Caregiver 3 (husband): "*Obviously, it's never as good as being there. From my perspective, I can do fine with this, I just feel bad for [patient's name] in particular, because as she's going into an appointment, especially if it's for the first time, at the cancer clinic for example, I can't imagine how intimidating that's got to be, when you go in there and you don't have anybody to be a second pair of ears. Because you can't have a phone with the consultation the whole way along, and you don't want to be that—I don't know—you don't want to be overbearing, but I just feel bad because [patient's name] is not able to have whatever support I might be able to be. Even when, she mentioned having surgery, and I couldn't go into the hospital with her, I couldn't check-in with her. So, I think that's a pretty—we understand the rationale, we don't begrudge that, it's just that it's a terrible time to be sick and as sick as my wife has been.*"

*Subtheme 2*: *Impact on relationships/ interactions with the health care team*. Participants felt that virtual visits (including telephone and video appointments) limited health care providers' ability to comprehensively assess a patient's current health status. Those who had a relationship with the oncology team prior to the pandemic or previous cancer treatment experiences were less likely to express concern about the absence of face-to-face appointments and reported that the pandemic had not impacted the relationship with their cancer treatment team, and they were generally satisfied with their cancer care. Older adults who were recently diagnosed for the first time with cancer who had never been treated prior to the pandemic, noted a lack of rapport with their health care provider due to a lack of face-to-face interactions and preferred to meet in person to receive the diagnosis to be able to develop a relationship with their oncology team but many were generally still satisfied with their cancer care. For some, the lack of face-to-face appointments led to gaps in information regarding treatment

and recovery plans. The importance of the ability to have clear and consistent communication with the health care team was highlighted as a major concern by patients and caregivers alike. No patient or caregiver had requested changes to the cancer treatment plan due to fear of COVID-19. Please see below for exemplar quotes:

> Patient 13 (female): *"Many times a phone call is fine, but there are times that I think that I know, or [caregiver's name] knows that I need to see the doctor and If it's your own family doctor, I think most times in the old days, you could say 'I need to see the doctor' and the secretary would find a way. Nowadays, if you need to see the doctor, God help you. The doctor doesn't know you and doesn't know what you need and, uhm, it's very difficult to be so estranged from your own doctor, and this doctor is very important now because I do have cancer and because it is COVID."*

> Patient 2 (male): *"I actually don't mind virtual as much as I do face-to-face. Often trying to understand someone who wears mask is difficult for me because [of] hearing loss and I also have tinnitus. It's very difficult to listen to somebody through the mask. So, I found the virtual is better for someone like that."*

> Caregiver 14 (daughter) and Patient 13 (female): *"So my mom and I are at home. She had the biopsy and a couple days later, we're in the laundry room and the phone rings and the oncologist says, 'Will you consent to talking on the phone?' and my mom said yes and then she says, 'Well, you have cancer,' and she starts kind of talking in language that we didn't really understand that and then she said, 'I've already put in orders for you to have chemo next week and I can't do surgery because normally I do,'—I don't know what she said, 18 or whatever a week- and now they've been cut down to two and you're not the highest priority, so [oncologist's name] can't offer surgery even though [oncologist's name] normally would and [patient's name] having chemo next week. That was it."*

**Theme 2: Navigating health and cancer systems in flux of the COVID-19 pandemic.** Within this theme we identified three subthemes: 1) Accessibility; 2) Impact on cancer diagnosis and treatment experienced; and 3) Implications of organizational changes in providing cancer care services.

*Subtheme 1*: *Accessibility*. As a result of the pandemic, many healthcare providers and specialists switched to virtual care, thereby limiting their ability to perform comprehensive physical assessments. Some of the patients interviewed stated difficulty navigating the technology, such as smart phones or tablets necessary for these virtual visits or required increased practical support from their caregiver. In addition, transportation to and from cancer treatments became challenging for caregivers due to 'no-visitor' policies implemented by health care facilities to reduce the spread of infection. As many of the caregivers interviewed expressed, older adults often have sensory impairments and lower levels of health literacy that effects their ability to understand their health situation, which then places the burden on caregivers to make sound treatment decisions on behalf of their loved ones. Some caregivers expressed frustration about not being able to join their loved one during the consultations and felt there should be exceptions for older adults to have their support person with them at their appointments. However, most older adults with cancer indicated that the consultations without their caregiver were manageable. None of the caregivers were invited to join by phone by the healthcare teams during the consultation appointment, but a few caregivers ensured they joined the telephone visits. Please see below for exemplar quotes.

Caregiver 4 (wife): *"So, this is why I want to go down with my husband, especially the first couple of times, so that I could hear what is being said, because you don't remember—I don't remember. It does help. I know they don't want really extra people in the hospital right now, and he's going down on Tuesday and [oncologist's name] had said I don't really need to come with him so I said, 'Well, that's fine,' I know he's going for some bloodwork, he's going for the CT thing, and he can drive himself so he'll be fine. But when he starts his radiation, I would like to maybe go at least maybe the first time, just to make sure he's okay when it's finished. He will be, but I'd just like to be there."*

Patient 3 (female): *"I would say for someone who is alone, and again, more elderly, or maybe doesn't understand English that well, or whatever the reason, they should have maybe a person that can help them plot out the calendar like I did, and as far as what is expected of me on this day and this day. Maybe if they could have even some volunteers help them do that—of course now with the pandemic you can't do that."*

Patient 13 (female): *"And it isn't only the older adults who sometimes can't deal with computers and deal with, uh, arranging appointments through computers. It's just there should be somewhere that if you have a choice, either you use the portal, or you can phone such and such a number and make it make an appointment. It [should] be like that."*

*Subtheme 2*: *Impact on cancer diagnosis and treatment experiences.* Although many of the patients and caregivers interviewed stated no significant impact on their or their loved one's treatment and none indicated that the treatment was reduced/shortened or otherwise modified due to COVID-19. However, some participants received their cancer diagnosis over the phone which they did not feel was the right way to receive a diagnosis, stressing the importance of face-to-face interactions with the healthcare team when discussing emotionally charged topics. Many participants reported that most had not explicitly discussed any changes to the treatment proposed to them because of the pandemic, and no patient had requested changes to their treatment out of fear of COVID-19. See below for exemplar quotes:

Patient 12 (male): *"It was mainly about the surgeries, you know that the surgeries were cancelled, heart surgeries were cancelled and obviously I connected it a little bit and I was concerned that [my surgery] would be delayed."*

Patient 13 (female): *"It's very difficult to be a patient during COVID, and as I said, the other day to my kids that that if it wasn't for COVID, I could fight cancer. Cancer isn't as hard as the combination is [it's] incredible. It's just impossible to deal with all the rules of COVID with mixed in with cancer and you have to follow the COVID rules. Never mind cancer. Your body tells you what cancer does to you, but. . . it's just impossible and people ignore the COVID rules. But I can't 'cause I'm too much in the medical system I have to abide by them and it's very, very difficult."*

*Subtheme 3*: *Implications of organizational changes in providing cancer care services.* Most patients expressed feeling safe whilst attending their hospital appointments in-person as they felt the hospital staff was very cautious and provided instructions on how to stay safe such as using physical distancing, masking, handwashing, etc. Most patients had a combination of telephone appointments and in-person appointments, while very few had video visits. Few indicated that public health guidelines such as masking, and distancing were not followed in the cancer centre leading to frustration. Some noted that there was a level of uncertainty regarding future treatment delays, specifically surgery postponements, which ultimately led to

worry and stress related to long-term cancer treatment outcomes such as survival. Others described feeling fortunate that they received timely treatments due to their own advocacy in tandem with their oncologist's support. See below for exemplar quotes.

> Patient 1 (male): *"It's uh, it's much easier sitting at home talking to yourself, and trying then for. . . getting downtown. . .getting in the hospital and then getting there and having to get back home again. That's 4 hours of your day and it's uh. . . it may not be a pain if you're only seeing the doctor once a year, but it's uh you have a [diagnosis] like I have and you are seeing doctors more often, uh virtual and phone is very effective."*

> Patient 11 (male): *"Oh do you know what I felt? I felt very safe. It's yeah, you know, I went to a number of hospitals before [hospital name] said, 'your hospital is very safe.' I didn't worry at all."*

The patients that engaged with the media surrounding the pandemic found that the misinformation and contradictory information related to COVID-19 death rates—as well as ageism—caused stress and impacted their ability to understand their own health situation in the context of their current need for treatment. Patients' and caregivers' who wanted to be vaccinated were frustrated by the delayed and differing provincial vaccination program rollouts and dose availability. Surprisingly, most study participants did not recall partaking in a discussion regarding the COVID-19 vaccine with their oncology team.

Few patients had recommendations regarding what needed to be improved in care during the ongoing pandemic. Some caregivers suggested that cancer centres ensure the support system of newly diagnosed cancer patients have adequate information on the possible impact of COVID-19 on their loved one's health and treatment. Another recommendation from caregivers was to ensure that older adults always have permission for their support system to join them for in-person appointments without the need to apply for special permission.

## Discussion

Our study aimed to increase the understanding of how the COVID-19 pandemic has affected the cancer treatment of older adults along with the experiences of their caregivers. Our most important finding is that older adults and their caregivers diagnosed during the pandemic experienced unique pressures related to treatment and decision making. These pressures were most evident in individuals who required surgery and were required to experience hospital stays alone. This has also been reported by [34, 35, 48]. Time limitations and visitation constraints around surgeries were also located within the context of ageist discourse related to resource allocation [53]. While participants were grateful for the care they received and that they were able to have their surgeries, the emotional impacts of experiencing these situations alone warrants further research about how these older adults and their caregivers can be better supported as visitor restrictions linger or return.

Individuals in this study were generally accepting of the treatment they received, experiencing only some inconveniences related to reliance on virtual care. This is consistent with other literature across the pandemic which indicates a willingness and openness from both older adults and clinicians on the reliance on virtual/telemedicine approaches [54]. A study assessing patient satisfaction with telemedicine in urology during the COVID-19 pandemic found that 83.8% of patients described the virtual visit as a good experience [55]. Similarly, the results of a recent study examining the perspectives of older adults receiving cancer care virtually revealed that 93% of their participants felt that the quality of the video

consultations was as good as face-to-face consultations and similar to having the specialist present with them in the room [41].

Although the use of telehealth provided flexibility and was easy to use for most, challenges were noted that need to be addressed. Some patients stated a difficulty establishing and maintaining rapport with new clinicians as a direct consequence of virtual appointments. While a few of the patients were apprehensive to experience this new method of providing care prior to meeting the specialist for the first time, most patients were ultimately accepting. Since the doctor-patient relationship is an important aspect of healthcare interactions, specialists need to ensure adequate time is spent to establish a rapport, rather than discussing medical matters with the patient immediately. It has been noted that the acceptance of teleconsultation appears to be linked to the patients' trust with their local health system and staff [41]. A survey among 241 patients with cancer in Italy [36] conducted during the first wave of the pandemic showed that 70% of patients with cancer agreed with the restrictions implemented in the cancer care unit to reduce the spread of COVID-19 including the switch to telephone appointments. However, almost a fifth of patients were not satisfied with follow-up and restaging visits completed by telephone [36]. Other studies note the importance of a pre-existing relationships between the patient and oncologist for the use of telehealth services [35, 48].

Some have also raised ethical concerns about the shift to telehealth. Research from France [37] amongst oncologists raised concerns about using telehealth due to their concerns about the suitability of the patient and disease. For example, will a patient with brain cancer, understand? As a result, the team developed a list of recommendations for whom telehealth is appropriate: *"1) the consultation is not the first consultation, 2) the patient and physician know each other and have a trust-relationship; 3) patient's physical examinations and evaluations must be good; 4) patient is on oral treatments/in post-treatment or surveillance phase; and 5) lives far away/unable to travel".* With ongoing pandemics and extended use of telehealth these guidelines can serve as guidelines for older adults to avoid experiences that patients in our study had with learning their diagnosis over the telephone from a physician with whom they had no relationship. These studies also show a need for training oncology staff on how to provide quality care through telehealth.

The benefits of telehealth lauded by participants which included reduced transportation time and attending appointments from their own home are approaches likely to be sustained in the future which were also mentioned by the oncologist in France [37]. Future exploration of how to maximize the potential of virtual health with older adults is an important area for future study and should take into account patient and caregiver preference. More research is needed to better understand what appointments could be held virtually without impacting patient-clinician relationships or understanding of the content of the consultation. Additional research is also needed to determine which patients need more support to use virtual care, and how caregivers can be involved in virtual care to keep them engaged in the decision-making process and treatment related information since they often provide the care at home. While there has been little reported on this, oncologists have also reported "missing half the team" in terms of information and providing instructions when caregivers are not able to join in person meetings due to COVID-19 restrictions [56, 57]. Going forward it is important to examine how caregivers can be involved in telehealth promoting support for the person with cancer and continuity of information for both the patient, caregiver and oncology team.

## Limitations

This is a qualitative study, and the intents of the study are not to generalize this data to other populations. Although an attempt was made to recruit both patients and caregivers from

varying sociodemographic backgrounds, most participants indicated that English was their first language and that they had attended College or University—as a result these findings should be interpreted cautiously. This study has several strengths, including the sampling approach in two diverse urban Canadian centres. As well, the qualitative approach allowed us to explore in depth the lived experiences of older adults with cancer and their caregivers, which would not be possible (or appropriate) with other methodologies.

## Conclusions

As the COVID-19 pandemic continues to disrupt our health care system's ability to conduct consultations in-person, the rapid uptake of virtual care was essential and mostly well-received. Our findings align with existing literature whilst providing new insight into how the pandemic has impacted the cancer treatment experiences of older adults and their caregivers. Although some older adults with cancer identified the impacts and pressures created by the pandemic on their treatment experiences, including added emotional stress and social isolation, many felt that the presence of COVID-19 did not impact treatment decisions made and available to them. The expansion of telehealth in the field of oncology will continue to change the way care and treatment is delivered; therefore, future studies should identify the best practices in the use of virtual care for the benefit of patients and their caregivers.

## Supporting information

**S1 Checklist. COnsolidated criteria for REporting Qualitative Research (COREQ) Checklist.**
(PDF)

**S1 File. Topic guides for older adults with cancer and caregivers.**
(DOCX)

## Acknowledgments

We would like to thank all participants for their time to participate in this interview.

## Author Contributions

**Conceptualization:** Caroline Mariano, Shabbir Alibhai, Manon Lemonde, Kristen Haase, Martine Puts.

**Data curation:** Isabel Tejero, Caroline Mariano, Shabbir Alibhai, Kristen Haase, Martine Puts.

**Formal analysis:** Cydney Low, Isabel Tejero, Nelly Toledano, Kristen Haase, Martine Puts.

**Methodology:** Caroline Mariano, Shabbir Alibhai, Manon Lemonde, Kristen Haase, Martine Puts.

**Project administration:** Martine Puts.

**Supervision:** Kristen Haase, Martine Puts.

**Writing – original draft:** Cydney Low, Kristen Haase, Martine Puts.

**Writing – review & editing:** Cydney Low, Isabel Tejero, Nelly Toledano, Caroline Mariano, Shabbir Alibhai, Manon Lemonde, Kristen Haase, Martine Puts.

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
