## [Decision Letter · Decision Letter 0]

9 May 2023

PONE-D-22-32970Impact of COVID-19 on older adults with cancer and their caregivers’ cancer treatment experiences study (ICE-OLD study)PLOS ONE

Dear Dr. Puts,

Thank you for submitting your manuscript to PLOS ONE. After careful consideration, we feel that it has merit but does not fully meet PLOS ONE’s publication criteria as it currently stands. Therefore, we invite you to submit a revised version of the manuscript that addresses the points raised during the review process.

We look forward to receiving your revised manuscript.

Kind regards,

Elisa Ambrosi

Academic Editor

PLOS ONE

2. Our staff editors have determined that your manuscript is likely within the scope of our Early Detection, Screening and Diagnosis of Cancer Call for Papers. This editorial initiative is headed by in-house PLOS editors. This Call for Papers aims to explore recent advances in the early detection of cancer and implications of these advances for patient survival. Additional information can be found on our announcement page: https://collections.plos.org/call-for-papers/early-detection-screening-and-diagnosis-of-cancer/

If you would like your manuscript to be considered for this collection, please let us know in your cover letter and we will ensure that your paper is treated as if you were responding to this call. Please note that being considered for the Call for Papers does not require additional peer review beyond the journal’s standard process and will not delay the publication of your manuscript if it is accepted by PLOS ONE. If you would prefer to remove your manuscript from collection consideration, please specify this in the cover letter.

“Dr. Puts is supported by a Canada Research Chair in the Care of Frail older adults.”

“The author(s) received no specific funding for this work”

Reviewers' comments:

Reviewer's Responses to Questions

**Comments to the Author**

1. Is the manuscript technically sound, and do the data support the conclusions?

Reviewer #1: Partly

Reviewer #2: Yes

2. Has the statistical analysis been performed appropriately and rigorously? 

Reviewer #1: Yes

Reviewer #2: N/A

3. Have the authors made all data underlying the findings in their manuscript fully available?

Reviewer #1: Yes

Reviewer #2: Yes

4. Is the manuscript presented in an intelligible fashion and written in standard English?

Reviewer #1: Yes

Reviewer #2: Yes

5. Review Comments to the Author

Reviewer #1: The authors have investigated the experiences of older patients with cancer and their caregivers during COVID-19 pandemic. This topic is particularly intriguing, considering the amount of data available for the general population comparing with the elderly (and their caregivers). Below some suggestions to improve the manuscript:

Overall:

Although the investigation covers an important issue, the central topic, i.e., COVID-19, may be unappealing since more and more papers have been published about the pandemic. Throughout the paper, the authors should highlight the novelty of their study and the potential implications for clinical practice.

Introduction:

- The authors should consider expanding the section dedicated to COVID-19, telemedicine, and healthcare organizational activities: The following references may address this issue:

https://pubmed.ncbi.nlm.nih.gov/34854782/

https://pubmed.ncbi.nlm.nih.gov/32580131/

- Page 2 line 2: please correct the typo “refesr”

Methods:

- The authors should specify the reference number of ethics committee approval, as well as if their study followed the Helsinki declaration and the principles of Good Clinical practice.

- The authors should consider specifying if specific guidelines for reporting this study (e.g., COREQ) have been followed.

- In the sample section, the authors should specify as the sample size has been determined. Usually, in qualitative research is applied the saturation principle since a priori sample size could not be calculated.

- In the recruitment and data collection, the authors should better specify how the interview guide has been developed. Was any theory (e.g., grounded theory, health belief model etc.) utilized to build the study and thus, the questions?

- In the analysis section, the authors should better explain how the thematic analysis was done. For instance, in qualitative research, the analysis is usually performed by two/three researches and then discussed together utilizing the triangulation system to resolve doubts and determine the final themes and sub-themes.

Results:

- In the description of the sample, the authors state that “eight caregivers and patients were interviewed together”. Was it a focus group?

- The themes are concise and well-written. Nevertheless, the authors should consider deleting the tables (3 and 4) and inserting some examples inside the results. Moreover, to facilitate the readers, the authors should consider dividing each theme into more sub-themes.

- The authors should consider inserting a figure to collect the major results that emerged from the study.

Discussion:

- Overall, the discussion could be expanded. More comparisons with other similar studies should be included.

- Again, to be more actual and appealing, the authors should highlight the potential implications for the practice of their research.

- Moreover, could the results also suggest that the use of telemedicine (e.g., the tools to perform telemedicine) should be personalized?

Reviewer #2: Thank you for the opportunity to review the manuscript entitled, “Impact of COVID-19 on older adults with cancer and their caregivers’ cancer treatment experiences study (ICE-OLD study)”.

In this multicentre qualitative study, 17 older adults with cancer and 10 caregivers were interviewed to elicit their experiences related to cancer treatment and caregiving in the context of the pandemic. The authors reported that older adults with cancer generally felt the pandemic did not impact treatment decision and access to care. Two key themes highlighted from thematic analysis are: alterations in individual and dyadic cancer experience and navigating health and cancer systems during the pandemic, and alluded to additional stressors brought about by the pandemic to older adults and their caregivers.

I would like to congratulate the authors on their study on this important topic, elucidating the perspectives of older adults and their caregiver in the context of oncology care and related experiences during the COVID-19 pandemic. The methodology and sampling strategy were appropriate; discussion is well-written, drawing on relevant, recent literature; and limitations are discussed. I imagine findings of this study will be valuable to guide approaches to care in future pandemics as well as in situations calling for augmented public health safety measures. I would like to offer some minor comments below for the authors’ consideration:

INTRODUCTION – Last sentence of 3rd paragraph: “Vulnerable populations in this context….” Please correct the spelling of ‘refers’ in this sentence.

INTRODUCTION – 4th paragraph: Would suggest adding transitional word (e.g. ‘Additionally’, or ‘Moreover’ to the beginning of the last sentence “These health measures post many challenges for older adults…..”.

METHODS - Given this is a qualitative study, the inclusion of the COREQ (COnsolidated criteria for REporting Qualitative research) Checklist in the reporting of findings would further strengthen this manuscript.

METHODS – Under Recruitment and data collection, last paragraph: “….and health status (approximately date of cancer diagnosis, self-reported cancer treatment….self-reported Charlson Comorbidity Index)….”. Would suggest replacing the wording ‘health status’ to ‘medical history and cancer-and treatment-related information’ to more appropriate reflect the type of information collected.

RESULTS – 4th line: “More than half of participants (58.8%) were in the age range of 70-80 years old.” Please rewrite as “More than half of the older adults participants (or patient-participants)….” to enhance clarity for this part, signifying you are referring only to the patients here.

RESULTS – Under Results of quality of study, 5th line: “…difficulty trying to visit their loved ones during admission.” To enhance clarity for readers, please specify for readers that this refers to ‘hospital admission’ as this is the first mention of ‘admission’ in this section.

Thank you again for the opportunity to review this important manuscript.

6. PLOS authors have the option to publish the peer review history of their article (what does this mean?). If published, this will include your full peer review and any attached files.

Reviewer #1: No

Reviewer #2: No

---

## [Author Response · Author response to Decision Letter 0]

21 Jul 2023

• Thank you for the links, we have updated the formatting as per the 2 linked guidelines. 

2. Our staff editors have determined that your manuscript is likely within the scope of our Early Detection, Screening and Diagnosis of Cancer Call for Papers. This editorial initiative is headed by in-house PLOS editors. This Call for Papers aims to explore recent advances in the early detection of cancer and implications of these advances for patient survival. Additional information can be found on our announcement page: https://collections.plos.org/call-for-papers/early-detection-screening-and-diagnosis-of-cancer/

If you would like your manuscript to be considered for this collection, please let us know in your cover letter and we will ensure that your paper is treated as if you were responding to this call. Please note that being considered for the Call for Papers does not require additional peer review beyond the journal’s standard process and will not delay the publication of your manuscript if it is accepted by PLOS ONE. If you would prefer to remove your manuscript from collection consideration, please specify this in the cover letter.

• Thank you for informing us of the call for the Early Detection, Screening and Diagnosis of Cancer Call for Papers, we would like to be considered for inclusion into this call. 

• Thank you for pointing out our omission on how consent was obtained. All participants provided written informed consent, and this has been added to the methods section in line 142 and in the box in the online submission system.

Text included: All older adults and caregivers provided written informed consent prior to the interviews. 

“Dr. Puts is supported by a Canada Research Chair in the Care of Frail older adults.”

“The author(s) received no specific funding for this work”.

• We apologize for including funding information in the manuscript. We have removed it from the manuscript and the sentence “The author(s) received no specific funding for this work” is correct. 

• The participants of this study did not give written consent for their data to be shared publicly, so due to the sensitive nature of the research supporting data is not available. Due to the nature of the interviews, we think the data cannot be de-identified further to be able to share them anonymously. 

Reviewers' comments:

Below we have addressed each concern raised point by point. To show the revised text we have included it below and included the line numbers that correspond to the line numbers in the version with tracked changes. 

Comments to the Author

1. Is the manuscript technically sound, and do the data support the conclusions?

Reviewer #1: Partly

Reviewer #2: Yes

2. Has the statistical analysis been performed appropriately and rigorously? 

Reviewer #1: Yes

Reviewer #2: N/A

3. Have the authors made all data underlying the findings in their manuscript fully available?

Reviewer #1: Yes

Reviewer #2: Yes

4. Is the manuscript presented in an intelligible fashion and written in standard English?

Reviewer #1: Yes

Reviewer #2: Yes

5. Review Comments to the Author

Reviewer #1: The authors have investigated the experiences of older patients with cancer and their caregivers during COVID-19 pandemic. This topic is particularly intriguing, considering the amount of data available for the general population comparing with the elderly (and their caregivers). Below some suggestions to improve the manuscript:

Overall:

Although the investigation covers an important issue, the central topic, i.e., COVID-19, may be unappealing since more and more papers have been published about the pandemic. Throughout the paper, the authors should highlight the novelty of their study and the potential implications for clinical practice.

Introduction:

- The authors should consider expanding the section dedicated to COVID-19, telemedicine, and healthcare organizational activities: The following references may address this issue:

https://pubmed.ncbi.nlm.nih.gov/34854782/ Exercise for counteracting post-acute COVID-19 syndrome in patients with cancer: an old but gold strategy?

https://pubmed.ncbi.nlm.nih.gov/32580131/ Organisational challenges, volumes of oncological activity and patients' perception during the severe acute respiratory syndrome coronavirus 2 epidemic

• Thank you for these suggestions, we have adjusted the introduction. We realize there have been more studies on the impact of COVID-19 on cancer care. Our analysis took more time than anticipated but we feel our study still adds value to the evidence as there have been few qualitative studies focusing on *older adults* and their treatment decision-making experiences, most studies have included younger populations who may have had different experiences as typically they are less reliant on caregivers, have less comorbidity and frailty and may have higher technology and health literacy skills as well as ageism may have impacted their experience differently. The purpose of this study was to study how care can be improved for this understudied population and that is relevant with virtual care to stay as well as for future pandemic outbreaks. We have not incorporated the first suggested reference; while very insightful in how the researchers used exercise in the case study of a patient with lung cancer with long covid, our study is not about the experience of cancer patients with long covid and the impact of the Covid-19 infection on their functional capacity. We have included the second reference in the introduction and discussion of this paper.

- Page 2 line 2: please correct the typo “refesr”

• Thank you for catching this typo. 

Methods:

- The authors should specify the reference number of ethics committee approval, as well as if their study followed the Helsinki declaration and the principles of Good Clinical practice.

• This study has been approved the University of Toronto Health Sciences Research ethics Board (#40167), the University Health Network research ethics board (#20-5948) and the UBC Harmonized ethics review board (H20-03957) and these numbers have been added in the text.

In Canada, research is guided by the Tri-Council Policy Statement Ethical Conduct for Research Involving Humans (https://ethics.gc.ca/eng/documents/tcps2-2022-en.pdf) which is based on the Helsinki declaration and the principals of Good Clinical practice. We have added the REB reference numbers and the tri council statement in the methods section. 

- The authors should consider specifying if specific guidelines for reporting this study (e.g., COREQ) have been followed.

• We have included the COREQ checklist as Supplemental file 2 and adjusted the manuscript to incorporate all checklist items. 

- In the sample section, the authors should specify as the sample size has been determined. Usually, in qualitative research is applied the saturation principle since a priori sample size could not be calculated.

• Thank you for pointing out this omission, we did indeed recruit until we noticed repetition in themes ad that which provided a robust responded to the research question. This has been added to the methods section. 

Text added in line 182/183: Recruitment continued until there was repetition in themes which robustly addressed the research questions. 

- In the recruitment and data collection, the authors should better specify how the interview guide has been developed. Was any theory (e.g., grounded theory, health belief model etc.) utilized to build the study and thus, the questions?

• The topic guide was developed by the research team which consisted of a medical oncologist, nurse researchers, geriatric oncologist and geriatricians involved in the care for older adults with cancer. The topic guide was not based on any particular model/theory as it was designed early in the pandemic as an exploratory study. The topic guide is appended in Supplemental file. The text has been clarified to indicate the research team developed the topic guide.

Text added line 194-197: Participants were interviewed using a topic guide developed by the multidisciplinary study team consisting of research nurses, a medical oncologist and geriatricians based on their experiences caring for older adults with cancer (see Supplemental file 1).

- In the analysis section, the authors should better explain how the thematic analysis was done. For instance, in qualitative research, the analysis is usually performed by two/three researchers and then discussed together utilizing the triangulation system to resolve doubts and determine the final themes and sub-themes.

• In this study, we involved three trainees who were relatively new to qualitative research (CL, NT and IT) and 2 experienced qualitative researchers (MP and KH). The entire team has coded all interviews and we held regular meetings to discuss the coded transcripts and themes and subthemes. This has been clarified in the manuscript.

Text added line 219-225: Five authors (all healthcare professionals with experience working with older adults with cancer) were directly involved in the thematic analysis (CL, IT, NT, MP, KH), they all coded all transcripts and then met on a regular basis to develop a coding framework and have discussions about fit and congruence across the developing themes and sub-themes. Three authors were trainees and relatively new to qualitative research (CL, IT and NT) and they were supervised by MP and KH who are experienced qualitative researchers.

Results:

- In the description of the sample, the authors state that “eight caregivers and patients were interviewed together”. Was it a focus group?

• No it were interviews with patients alone or interviews with the dyad of patient-caregiver. It has been clarified in the methods section. 

Text added line 193/194: For participants whose caregiver agreed to be included in the study, the patient and caregiver could choose to do the interview together as a dyad or each an individual interview.

- The themes are concise and well-written. Nevertheless, the authors should consider deleting the tables (3 and 4) and inserting some examples inside the results. Moreover, to facilitate the readers, the authors should consider dividing each theme into more sub-themes.

• Thank you for your kind feedback. We have removed the tables and inserted the quotes in the results section. We have not re-analyzed our data to come up with more subthemes. Our team of researchers came up with these themes and subthemes after all our coding and we feel we cannot suddenly add additional subthemes after our analysis. 

- The authors should consider inserting a figure to collect the major results that emerged from the study.

• Thank you for that suggestion, we have created a figure to highlight the major results. 

Discussion:

- Overall, the discussion could be expanded. More comparisons with other similar studies should be included.

• Thank you for that suggestion, we have expanded the discussion to add recent similar studies on experiences of other cancer patients with accessing cancer care, using telehealth, the impact on the patient-physician relationship and the change in the ability of caregivers in the consultation (lines 519-552). 

- Again, to be more actual and appealing, the authors should highlight the potential implications for the practice of their research.

• Thank you for your suggestion, we included potential implications for research. 

Text added line 526: More research is needed to better understand what appointments could be held virtually without impacting patient-clinician relationships or understanding of the content of the consultation. Additional research is also needed to determine which patients need more support to use virtual care, and how caregivers can be involved in virtual care to keep them engaged in the decision-making process and treatment related information since they often provide the care at home.

- Moreover, could the results also suggest that the use of telemedicine (e.g., the tools to perform telemedicine) should be personalized?

• We have expanded on our discussion on telemedicine in the discussion as well as the need to further study how to optimize the use of telehealth and include caregivers in the virtual consultations (line 538-552). 

Reviewer #2: Thank you for the opportunity to review the manuscript entitled, “Impact of COVID-19 on older adults with cancer and their caregivers’ cancer treatment experiences study (ICE-OLD study)”.

In this multicentre qualitative study, 17 older adults with cancer and 10 caregivers were interviewed to elicit their experiences related to cancer treatment and caregiving in the context of the pandemic. The authors reported that older adults with cancer generally felt the pandemic did not impact treatment decision and access to care. Two key themes highlighted from thematic analysis are: alterations in individual and dyadic cancer experience and navigating health and cancer systems during the pandemic, and alluded to additional stressors brought about by the pandemic to older adults and their caregivers.

I would like to congratulate the authors on their study on this important topic, elucidating the perspectives of older adults and their caregiver in the context of oncology care and related experiences during the COVID-19 pandemic. The methodology and sampling strategy were appropriate; discussion is well-written, drawing on relevant, recent literature; and limitations are discussed. I imagine findings of this study will be valuable to guide approaches to care in future pandemics as well as in situations calling for augmented public health safety measures.

• Thank you for your kind words! That is exactly what we had in mind. We (really) appreciated your formulation, so we have included that in our introduction. 

 I would like to offer some minor comments below for the authors’ consideration:

INTRODUCTION – Last sentence of 3rd paragraph: “Vulnerable populations in this context….” Please correct the spelling of ‘refers’ in this sentence.

• Thank you, we have corrected this typo. 

INTRODUCTION – 4th paragraph: Would suggest adding transitional word (e.g. ‘Additionally’, or ‘Moreover’ to the beginning of the last sentence “These health measures post many challenges for older adults…..”.

• Thank you, we added “Moreover” to the suggested sentence.

METHODS - Given this is a qualitative study, the inclusion of the COREQ (COnsolidated criteria for REporting Qualitative research) Checklist in the reporting of findings would further strengthen this manuscript.

• Thank you, the COREQ checklist has been included in supplemental file 2. 

METHODS – Under Recruitment and data collection, last paragraph: “….and health status (approximately date of cancer diagnosis, self-reported cancer treatment….self-reported Charlson Comorbidity Index)….”. Would suggest replacing the wording ‘health status’ to ‘medical history and cancer-and treatment-related information’ to more appropriate reflect the type of information collected.

• Thank you, we have changed the wording to the suggested wording.

RESULTS – 4th line: “More than half of participants (58.8%) were in the age range of 70-80 years old.” Please rewrite as “More than half of the older adults participants (or patient-participants)….” to enhance clarity for this part, signifying you are referring only to the patients here.

• Thank you, we have adjusted this wording to the suggested wording. 

RESULTS – Under Results of quality of study, 5th line: “…difficulty trying to visit their loved ones during admission.” To enhance clarity for readers, please specify for readers that this refers to ‘hospital admission’ as this is the first mention of ‘admission’ in this section.

Thank you again for the opportunity to review this important manuscript.

• Thank you, we have added hospital to clarify that it was a hospital admission.

---

## [Decision Letter · Decision Letter 1]

5 Sep 2023

Impact of COVID-19 on older adults with cancer and their caregivers’ cancer treatment experiences study (ICE-OLD study)

PONE-D-22-32970R1

Dear Dr. Puts,

We’re pleased to inform you that your manuscript has been judged scientifically suitable for publication and will be formally accepted for publication once it meets all outstanding technical requirements.

Kind regards,

Elisa Ambrosi

Academic Editor

PLOS ONE

Additional Editor Comments (optional):

Reviewers' comments:

Reviewer's Responses to Questions

**Comments to the Author**

1. If the authors have adequately addressed your comments raised in a previous round of review and you feel that this manuscript is now acceptable for publication, you may indicate that here to bypass the “Comments to the Author” section, enter your conflict of interest statement in the “Confidential to Editor” section, and submit your "Accept" recommendation.

Reviewer #1: All comments have been addressed

Reviewer #2: All comments have been addressed

2. Is the manuscript technically sound, and do the data support the conclusions?

Reviewer #1: Yes

Reviewer #2: Yes

3. Has the statistical analysis been performed appropriately and rigorously? 

Reviewer #1: N/A

Reviewer #2: N/A

4. Have the authors made all data underlying the findings in their manuscript fully available?

Reviewer #1: Yes

Reviewer #2: Yes

5. Is the manuscript presented in an intelligible fashion and written in standard English?

Reviewer #1: Yes

Reviewer #2: Yes

6. Review Comments to the Author

Reviewer #1: We thank the authors for revising the manuscript according to our comments. The manuscript is now substantially improved.

Reviewer #2: (No Response)

7. PLOS authors have the option to publish the peer review history of their article (what does this mean?). If published, this will include your full peer review and any attached files.

Reviewer #1: No

Reviewer #2: No

---

## [Editor Report · Acceptance letter]

12 Sep 2023

PONE-D-22-32970R1 

Impact of COVID-19 on older adults with cancer and their caregivers’ cancer treatment experiences study: the ICE-OLD study 

Dear Dr. Puts:

I'm pleased to inform you that your manuscript has been deemed suitable for publication in PLOS ONE. Congratulations! Your manuscript is now with our production department. 

Kind regards, 

on behalf of

Dr. Elisa Ambrosi 

Academic Editor

PLOS ONE